# Adoption of Sustainable Agriculture Intensification in Maize-Based Farming Systems of Katete District in Zambia

Petan Hamazakaza [1,*], Gillian Kabwe [1], Elias Kuntashula [2], Anthony Egeru [3] and Robert Asiimwe [4]

1   School of Natural Resources, Copperbelt University, Kitwe P.O. Box 21692, Zambia; gillian.kabwe@cbu.ac.zm
2   Department of Agricultural Economics and Extension, University of Zambia, Lusaka P.O. Box 32379, Zambia; ekuntashula@unza.zm
3   Department of Environmental Management, Makerere University, Wandegeya, Kampala P.O Box 16811, Uganda; a.egeru@ruforum.org
4   Department of Agribusiness and Natural Resource Economics, Makerere University, Wandegeya, Kampala P.O Box 16811, Uganda; r.asiimwe@cgiar.org
*   Correspondence: hamazakazap@yahoo.com; Tel.: +260-977-440948

**Abstract:** Sustainable agricultural intensification (SAI) has been hailed as the solution to increasing crop productivity among farmers. Despite the significant promotion, there still remains a dearth of information on the adoption and intensity of SAI in Zambia. This study sought to identify factors that influence farmers' adoption of SAI practices and intensity of use. A cross-sectional survey was conducted among 300 smallholder farmers of Katete district in Zambia. The Cragg's double hurdle model was used to assess the key decision factors for SAI adoption and intensity of use. Empirical estimates revealed that limited years of farming and smaller total cropped field size were statistically significant decision factors that led to a reduced likelihood of SAI adoption. The results of the truncated model showed that smaller farm sizes and limited access to farmer extension services reduced the adoption intensity of SAI practices, whereas farmer affiliation with farmer associations and farmer training in crop production increased SAI adoption intensity. We recommend an increase in farmer training on and sensitization to the benefits of SAI practices aligned to their respective landholdings.

**Keywords:** sustainable agriculture intensification; adoption; double hurdle; Katete; Zambia



## 1. Introduction

Agriculture remains the most important sector in Sub-Saharan Africa (SSA) as a source of livelihood and employment for about 65% of the region's labor force, yet it is still under-utilized far below its potential, with gains in land and labor productivity lagging behind those of other regions [1]. It is becoming more difficult nowadays for many SSA countries to realize agricultural growth by putting more land under cultivation [2]. Productivity growth is essential to satisfying the growing food demands and poverty alleviation in developing countries, though future productivity growth also hinges on preserving natural resources [3]. The agricultural sector in SSA is marked by low productivity with little application of science and technology to help sustain soil productivity. The region accounts for about 22% of the total global cost of land degradation, which is currently estimated at USD 300 billion [2]. Most of the soil degradation and, in particular, soil fertility depletion in SSA is associated with the use of unsustainable practices in the production of economically important crops that result in farmers' continued engagement in deforestation and forest degradation practices in efforts to maintain some economic level of crop yield [3]. As a result, lower crop yields and a large yield gap have remained pervasive across SSA. Identifying ways to improve land productivity and sustainability for smallholder farmers who are often most negatively affected by land degradation is a central challenge in reversing the declining per capita food availability [4]. Other scholars [1] have recommended

that the successful use of agriculture as an instrument for poverty reduction and meeting food security while conserving the environment in SSA requires greater attention from governments and donors, supported by scholarship and learning. Recent studies [5,6] have shown that SSA has been experiencing an increase in cereal production. However, these gains are largely a result of agricultural extensification (increasing the land cleared for cultivation), leading to land degeneration.

The soil fertility challenge, increasingly emerging from intensifying cultivation without planned replacement of depleted soil nutrients, remains critical across SSA. As a result, it has become a major impediment to higher production among farmers in the region. Many actions, particularly enhanced farmer training, facilitation of farmers' access to appropriate inputs for agricultural production, policy engagement, and support to farmers can be taken to reverse the degradation trend. Sustainable land management (SLM) practices such as those centered on prevention of land conversion, protection of vulnerable lands, management, and enhancement of soil fertility offer a set of solutions that are generally acceptable [7]. However, even when the potential to benefit from SAI practices exists, accessing the information on the sources of inputs and other associated services required for the adoption of SAI practices can be a significant limitation for the millions of smallholders in Sub-Saharan Africa [8]. This notwithstanding, the attributed benefits of SAI have left it as one of the most preferred approaches for addressing the agricultural production challenges of resource-poor farmers in SSA [9].

The crop production sector, particularly maize cultivation, is the dominant source of livelihood for small-scale farming in Zambia [10]. Agriculture employs 72% of the country's workforce, and rural areas are home to more than 60% of the population [11]. Zambia is the third-hungriest country in the world, after Chad and the Central African Republic, according to the 2016 *Global Hunger Index* report (GHI), suggesting severe levels of hunger [12]. In this context, agricultural development is seen as a key to combating food insecurity and high poverty levels among rural farm households [13]. Past studies [14] report that to sustain increased food security, it is critical to maintain soil fertility and higher food productivity with respect to environmental challenges. Furthermore, [15] reiterates the critical role of adopting environmentally friendly sustainable agricultural intensification practices in maintaining soil fertility for sustained crop productivity. Various SAI practices have been discussed to promote crop yields and environmental conservation. However, in this article, we analyze the adoption of SAI practices that are relevant to the Zambian situation, such as growing drought, pest and disease stress-tolerant crops, use of minimum tillage, green manure incorporation, integrated livestock-crop production, precision fertilizer application, agroforestry, kraal manure, crop rotation, crop diversification, cover crops, intercropping, and crop variety diversification. The challenges to increasing crop productivity in Zambia are, in part, related to the decline in soil fertility due to the low adoption of SAI practices [10].

Farmers affected by soil fertility depletion in Zambia (especially those within traditional land tenure systems) often resort to shifting cultivation, clearing, and opening new land as a copping strategy. These actions lead to short-term gains in production but with considerably high rates of deforestation in the country, leading to land degradation. Reports indicate that between 2000 and 2014, the national estimated total deforestation rate in Zambia varied between 250,000 and 300,000 hectares per year, representing an annual deforestation rate of 0.7% of the forestry cover, which stood at 45 million hectares [16]. During the same period, around 156,000 hectacres of forest were believed to have been destroyed in the Eastern Province, mainly owing to agricultural expansion [16,17]. Thus, by opening up new land, extensive land-use change has been registered as well as resultant land degradation due to land use and land cover change. The costs of this degradation are most felt among poor smallholder farmers. It is now generally appreciated that the cost of taking action against land degradation is much lower than the cost of inaction and the returns to taking action are higher [3].

Sustainable agriculture intensification (SAI) has been proposed as one approach that has the potential to contribute to reconciling the need for more environmentally benign agriculture while advancing global food security [18]. Empirical evidence [19,20] demonstrates that a win–win relationship exists between sustainable agricultural intensification and forest conservation. Sustainable agriculture intensification aims to increase food production from existing farmland while avoiding negative environmental, social, or cultural impacts [21,22]. SAI has the potential to increase food security without detrimental effects on ecosystem services. Despite its apparent vital contributions, SAI, like other sustainable land use management practices, has had some discourse pertaining to its efficiency in addressing the challenges of land degradation while limiting agricultural expansion [23]. It is estimated that the adoption of sustainable land management practices in the SSA region has been very low, covering barely 3% of total cropland [24]. After years of SAI promotion in Zambia, adoption levels of SAI practices remain low even among the target populations. Constraints identified by previous studies have presented concerns of suboptimal understanding of trade-offs and synergies among target users [25]. Further, information asymmetries at the farm level have constituted barriers to adoption among resource-poor farmers [26]. These challenges may not be any different in the context of Zambia, but studies so far undertaken have focused on the review of agronomic practices and their benefits [27,28], and impacts of SAI in terms of enhancing productivity, product diversity, food security, and land tenure implications on adoption [10,29]. Accordingly, the depth of adoption drivers and intensity of SAI application remain largely explored. Therefore, addressing the aspects of sustainable agricultural practices, particularly understanding the adoption behavior of farmers in more depth, is imperative to attain sustained agricultural productivity and food security status of smallholder farms [29]. This study sought to determine the factors that influence farmers' decisions to adopt and the level of adoption of SAI in rural smallholder farmer communities of Eastern Zambia.

## 2. Theoretical Overview of Technology Acceptance and Use

### 2.1. Unified Theory of Acceptance and Use of Technology

Based on a comprehensive review and synthesis of several theoretical models, some scholars [30] proposed the unified theory of acceptance and use of technology (UTAUT), which has since been used extensively by researchers in their quest to explain technology acceptance and use. The original UTAUT model explained a considerable amount of variance in behavioral intention and usage behavior. This theory indicates that farmers' decisions to use sustainable agricultural production methods are influenced by factors such as perceived benefits, ease of application compared to capacity, the influence of society on the role of agricultural extension activities, the availability of resources such as human capital, physical capital, accessibility to other resources, and household demographic characteristics.

### 2.2. Technology Acceptance Theory

When it comes to new technology, other researchers [31] believe that there is a causal relationship between its usefulness and users' sentiments. Perceived usefulness influences user attitudes, and perceived ease of use is the degree to which consumers believe it is simple to use when put into reality. Farmers will choose to use sustainable agricultural production methods, according to this theory, if they are confident in the benefits that these measures provide and have the ability to put these measures into practice without encountering too many barriers in terms of knowledge and resources.

### 2.3. Theory of Technology Dissemination

Some studies [32] address the impact of two elements on the adoption of a new technology: compatibility and benefits. This theory describes the five processes through which ideas and technology are diffused and accepted: awareness, persuasion, decision-making, implementation, and validation. Accepting technology selection is influenced by a number of criteria, including linked benefits, flexibility, ease of access, ease of testing, and ease of

observation. The theory suggests that when transferring new agricultural technologies, the extension system should pay attention to farmers' decision-making processes.

### 2.4. Theory of Planned Behavior

According to the literature [33], the theory of planned behavior (TPB) has three elements that influence behavior: attitude toward the behavior, subjective criteria, and behavioral control awareness. As a result of its origins in the notion of rational action, the TPB overcomes the constraint that human conduct cannot be entirely controlled. In this approach, there are three basic determinants: the personal factor is the individual's attitude toward the behavior in terms of positive or negative awareness of performing the behavior, and the subjective norm is the determinant of self-efficacy or the ability to perform a behavior because it copes with the perception of pressure or subjective compulsion, and finally, the behavioral norm is the determinant of self-efficacy or the ability to perform a behavior because it copes with the perception of pressure or subjective compulsion.

The above theories are relevant to this study, where it is important to explain the factors that influence farmer decisions, the role of various factors in influencing farmer behavior with respect to technology and practice adoption, and perceived usefulness in influencing behavior decisions to apply new technology and technical solutions.

## 3. Materials and Methods

### 3.1. Study Area

This study was conducted in two sites, namely, Vulamukoko and Lukweta communities which are respectively, located in Vulamukoko and Chimtende Agricultural Camps of Katete district in the Eastern province of Zambia (Figure 1). Katete district has a population of 1.7 million people of which the majority (87%) live in rural areas [34]. This population's livelihood is dependent on agriculture, especially crop and livestock production, and natural resource utilization. The district is located at 32.0440° E and 14.0584° S, and stands at an altitude of 1060 m.a.s.l. These sites were purposively selected to represent the communities close to forest-protected areas (Lukweta) and those that are far from a forest-protected area (Vulamukoko). The Lukweta area borders the Chindindendi and Mulodzela protected forest areas that farmers have encroached on for farming activities. The Vulamukoko area is approximately 30 km from the nearest forest-protected area.

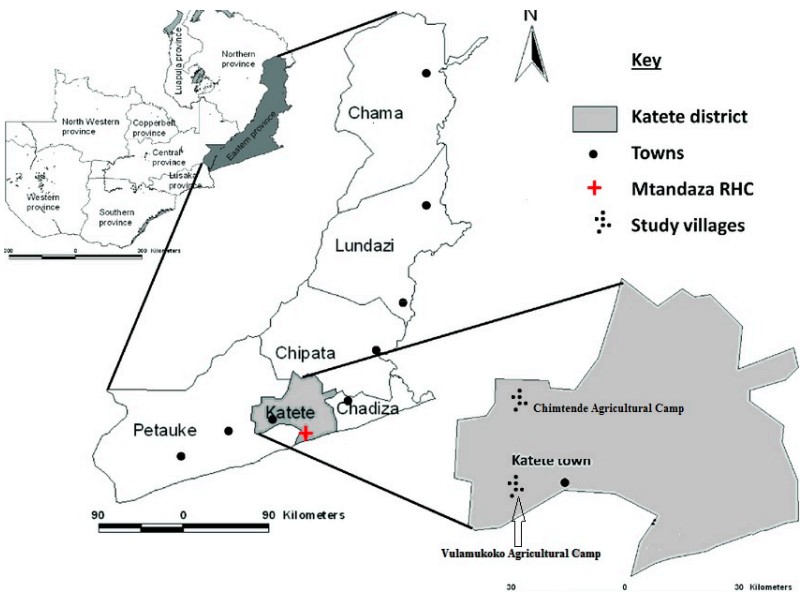

**Figure 1.** Location of Katete district in Zambia. Source: Authors' compilation.

### 3.2. Household Sampling

Farm households were selected using a two-stage sampling technique. The Lukweta community has 147 villages with an estimated farm household population of more than 6500 families. The Vulamukoko area has 58 villages with approximately 3007 farm families. The number of households and household lists in each village were obtained from the Chimtende and Vulamukoko Agricultural Camp Farmer Registers for the year 2020. Using this data, the first stage of sampling involved the selection of five villages, each from Lukweta and Vulamukoko. The selected villages in the Lukweta community were Jamani, Mbalani, Mulipa, Chimbwala, and Cafela, which comprised 145, 120, 95, 105, and 115 households, respectively. In the Vulamukoko community, the selected villages were Chapita, Kazika, Azeleguze, Kawalala, and Azelekaceka, which comprised 189, 130, 176, 200, and 145 households, respectively.

In the second sampling stage, the probability proportionate to size (PPS) sampling method was used to determine the number of households to be sampled from each of the selected villages. Village size was measured as the number of households in each village. PPS controls for differences in the size of villages. A systematic random sampling approach was used to sample households in each village. In total, 300 households (150 from each community) representing 21.1% of the total household population in the area were selected based on the available financial resources.

### 3.3. Data and Data Collection

A pre-tested semi-structured questionnaire was used to collect data through face-to-face interviews with the help of two well-trained research assistants. The questionnaire was administered at the household level and captured information on household demographic characteristics, capital assets, land characteristics and distribution, agricultural expansion, household organizational affiliation and training, and sustainable agriculture intensification practices. The survey was conducted from January–April 2021.

### 3.4. Econometric Analysis

This study followed the Smith theory of dependency in double-hurdle models. Smith's theory was designed to explain individual demand through a simultaneous two-step process of a market participation decision (first hurdle), and a consumption-level decision (second hurdle), where a non-zero correlation/covariance parameter allows for dependency between the hurdles. This theory has been extended to farmer technology adoption in past studies [35]. In the first hurdle, the farmer is faced with the decision to adopt or not adopt SAI [35]. Holding all other factors constant, such as access to information, access to production resources, farmer demographic and socioeconomic attributes, and physical farm characteristics, it is assumed that a farmer will choose to adopt SAI if the benefits from adoption outweigh the benefits of non-adoption. In the second hurdle, a farmer has to decide how much field area to cultivate under the SAI practices chosen to be adopted from the first decision hurdle. The decision model is taken as a two-step process and considers that there might be a possibility of a zero observation in the first step, which may arise from the farmer's decision whether or not to adopt.

Since the intensity of SAI adoption measured as the area of land cultivated under SAI practices takes on positive values censored at zero (for non-adopters), a tobit model could be used to analyze both adoption and intensity [36]. However, the tobit approach assumes that the decision to adopt and the decision relating to intensity are the same, which may not be appropriate [36,37]. In this study, these decisions were treated as separate processes because the farmer must first make a decision to use SAI practices and then decide on the rate and extent of their use in each season. Moreover, factors that affect the decision to adopt SAI practices may differ from those that affect the intensity of their application [37]. Based on Smith's theory and the stated reasons for the inapplicability of the tobit in this study, the double hurdle model was applied to address two farmer decision levels centered

on the decision to adopt (allocate resources) or not to adopt SAI as the first hurdle, and how intensely (amount of resources to allocate) as the second hurdle [36].

The first hurdle of the double hurdle model estimation uses a probit regression, which takes 0 as the decision not to use SAI practices, and 1 as the decision to use regardless of how much is applied. This can be specified as in Equation (1).

$$\rho\left(w = \frac{1}{x}\right) = \varphi(x\gamma) \tag{1}$$

where $\rho$ denotes the probability of adoption of SAI practices, $w$ is a binary variable of SAI adoption decision (adopt or not adopt), $\varphi$ represents the cumulative normal distribution of the adoption decision in the study population, $x$ is a vector of farm and household characteristics that may influence adoption, and $\gamma$ represents the vector of coefficients to be estimated.

The second hurdle uses a truncated regression model to determine the factors that explain the intensity of adoption for the subset of individuals who adopt [38] as specified in Equation (2).

$$y = x\beta + \varepsilon \tag{2}$$

where $y$ represents the vector of SAI intensity levels, $x$ is a vector of farm and household characteristics that may influence intensity levels, $\beta$ is a vector of estimated parameters, and $\varepsilon$ is a vector of the error term.

Observed factors that were expected to affect these decisions included several household and farm characteristics considered by other researchers [39–43]. In particular, this analysis included the following variables in the model: farm household size, number of economically active adults, age of farmer, years of farmer experience in farming, household head ability to read and write, number of work oxen owned, the total size of the farmland, total cropped field size, affiliation of household member to farmer associations, farmer access to agricultural extension training and type of extension training received.

## 4. Results

### 4.1. Sociodemographic Characteristics of Respondents

Results showed that 61.5% of the sampled households were utilizing SAI practices. Table 1 compares adopters and non-adopters of SAI practices based on socioeconomic and demographic household characteristics in the two study communities. The comparisons reveal a more homogenous community in Vulamukoko community with adopting and non-adopting households only differing in terms of the average number of work oxen owned, household members belonging to farmer groups/cooperatives, and household members having benefited from training in the past 5 years prior to the study. On the other hand, adopting and non-adopting households in Lukweta community were different in human capital (literacy), farming resources endowment (total farmland owned), production levels (total farmland cultivated), and institutional setup (affiliation to farmer associations, access to agricultural extension training and the types of training farmers received) factors.

**Table 1.** Socioeconomic and demographic characteristics of the study households, by SAI adoption.

| Household Characteristics | Lukweta Community | | | Vulamukoko Community | | |
|---|---|---|---|---|---|---|
| | SAI Adopters (*n* = 63) | SAI Non-Adopters (*n* = 93) | | SAI Adopters (*n* = 76) | SAI Non-Adopters (*n* = 16) | |
| | Mean (Std. Dev.) | Mean (Std. Dev.) | *p*-Value | Mean (Std. Dev.) | Mean (Std. Dev.) | *p*-Value |
| Household size | 6.825 (2.2471) | 6.696 (2.7843) | 0.759 | 6.303 (2.0332) | 6.063 (2.5682) | 0.683 |

**Table 1.** *Cont.*

| Household Characteristics | Lukweta Community | | | Vulamukoko Community | | |
|---|---|---|---|---|---|---|
| | SAI Adopters (*n* = 63) | SAI Non-Adopters (*n* = 93) | | SAI Adopters (*n* = 76) | SAI Non-Adopters (*n* = 16) | |
| | Mean (Std. Dev.) | Mean (Std. Dev.) | *p*-Value | Mean (Std. Dev.) | Mean (Std. Dev.) | *p*-Value |
| Household members aged 16–59 years | 2.921 (1.4176) | 3.151 (1.6936) | 0.376 | 2.868 (1.4174) | 2.625 (1.1475) | 0.522 |
| Age of household head (years) | 43.02 (11.508) | 43.86 (12.929) | 0.676 | 44.50 (10.961) | 45.31 (14.449) | 0.800 |
| Household head years of farming | 21.38 (10.379) | 21.59 (11.570) | 0.908 | 20.03 (12.055) | 14.50 (13.962) | 0.108 |
| Household head ability to read and write (1 = yes; 0 otherwise) | 0.84 (0.368) | 0.69 (0.466) | 0.030 | 0076 (0.428) | 0.75 (0.447) | 0.912 |
| Number of work oxen owned | 2.66 (1.948) | 2.59 (1.219) | 0.827 | 2.69 (1.451) | 1.43 (0.535) | 0.029 |
| Total farmland owned (ha) | 7.29 (7.558) | 5.27 (2.918) | 0.023 | 3.93 (2.782) | 3.13 (1.695) | 0.292 |
| Total farmland cultivated (ha) | 5.37 (2.878) | 4.23 (1.931) | 0.003 | 4.05 (2.429) | 3.58 (1.564) | 0.462 |
| Household members belonging to farmer groups/cooperatives (1 = yes; 0 otherwise) | 0.81 (0.396) | 0.49 (0.503) | 0.000 | 1.00 (0.000) | 0.81 (0.403) | 0.000 |
| Has any household member benefited from training in the past 5 years | 0.79 (0.408) | 0.45 (0.500) | 0.000 | 0.92 (0.271) | 0.75 (0.447) | 0.046 |
| Household head received crop production training (1 = yes; 0 otherwise) | 1.00 (0.000) | 1.26 (0.677) | 0.010 | 1.00 (0.000) | 1.00 (0.000) | |
| Percent SAI adoption | 40.4 | 59.6 | | 82.6 | 17.4 | |

Source: Authors. *p*-values refer to independent-samples two-tailed *t*-tests.

### 4.2. Level of Household Adoption of Sustainable Agricultural Intensification Practices

The results showed that 56% of the sampled households were using SAI practices on about 0.8 ha of land per household. As presented in Table 2, the most practiced SAI techniques were crop rotation and having well-designated grazing areas (78%), followed by minimum tillage (65%). The least applied SAI practices were legume crop non-incorporation soil fertility practices (16%), green manure incorporation (19%), and cover crops (19%). The low application of these practices is mainly attributable to seed availability challenges, particularly for non-edible legume species. Using the principal components analysis of the SAI practices that are being promoted in the study area and the country as a whole, farmers were classified into three clusters. These were centered on improved agronomic practices, legume-based soil fertility management practices, and crop-livestock integration practices. Clustering of the SAI practices was based on average percentages in terms of the adoption of SAI practices, where improved agronomic practices were the most practiced, followed by legume-based soil fertility management practices, and the least being crop–livestock integration practices.

**Table 2.** Classification of SAI practices based on Principal Components Analysis.

| Variable | Adoption (%) | Improved Agronomic Practices | Legume-Based Soil Fertility Management Practices | Crop–Livestock Integration | Communality |
|---|---|---|---|---|---|
| Grow drought, pest, and disease stress-tolerant crops | 53% | 0.83 | 0.24 | 0.05 | 0.74 |
| Use minimum tillage | 65% | 0.76 | 0.10 | 0.30 | 0.68 |
| Green manure incorporation | 19% | 0.32 | 0.36 | 0.02 | 0.23 |
| Integrated livestock–crop production | 28% | 0.26 | 0.24 | 0.29 | 0.21 |
| Precision fertilizer application | 41% | 0.75 | 0.37 | −0.03 | 0.70 |
| Agroforestry soil fertility practices | 35% | 0.56 | 0.35 | 0.01 | 0.44 |
| Kraal manure soil fertility practices | 43% | 0.49 | 0.48 | 0.22 | 0.52 |
| Green manure non-incorporation soil fertility practices | 16% | 0.21 | 0.71 | 0.02 | 0.56 |
| Crop rotation | 78% | 0.44 | 0.09 | 0.61 | 0.57 |
| Integrated pest management methods | 57% | 0.82 | 0.15 | 0.15 | 0.72 |
| Crop diversification | 54% | 0.87 | 0.14 | 0.10 | 0.79 |
| Cover crops | 19% | 0.23 | 0.79 | 0.07 | 0.69 |
| Intercropping | 61% | 0.29 | 0.64 | 0.11 | 0.50 |
| Extension trainings | 57% | 0.74 | 0.20 | 0.13 | 0.61 |
| Herbicide weed control | 53% | 0.76 | 0.18 | 0.15 | 0.63 |
| Crop variety diversification | 42% | 0.88 | 0.21 | 0.06 | 0.82 |
| Designated grazing areas | 78% | −0.10 | 0.08 | 0.74 | 0.56 |
| Eigenvalues | | 8.13 | 1.39 | 1.17 | |
| Eigenvalues % contribution | | 76.05 | 12.97 | 10.99 | |
| Cumulative % | | 76.05 | 89.02 | 100.00 | |

Table 2 presents principal components (PCs) and the coefficients of linear combinations called loadings. The results presented in Table 2 revealed a good fit, indicating that the factor analysis results greatly explained the data. The first component was found to explain 76.05% of the variance and is correlated with the growing of drought, pest, and disease stress-tolerant crops, the use of minimum tillage, precision fertilizer application, agroforestry soil fertility practices, integrated pest management methods, crop diversification, extension training, herbicide weed control, and crop variety diversification, all with high positive effects (factor loadings). Thus, this component was named "improved agronomic practices". Principal components 2 and 3 accounted for 12.97% and 10.99% of variances, respectively. This means that the first three components are of greater importance in explaining the variation in the dataset. The second PC was associated with green manure non-incorporation soil fertility practices, cover crops, and intercropping, all with positive loadings. The second component was thus named "legume-based soil fertility management practices". The third PC was associated with crop rotation and designation of livestock grazing areas, all with positive effects (loadings). As such, this component was named "crop–livestock integration". The communality column presents the total amount of variance of each variable retained in the three components, with an average communality of 60% for the entire sample. Additionally, the Kaiser–Meyer–Olkin (KMO) statistic was 0.9, which is above the recommended 0.5 to justify sampling adequacy, thus affirming principal component analysis as an appropriate method.

Factors Determining Adoption, Intensity, and Extent of SAI Use

The double hurdle regression results are presented in Table 3. The regression model's Wald statistic was significant at 1%, suggesting a good fit of the model as a whole. The first hurdle shows the factors that influence the decision to use SAI components, while the second hurdle shows factors that influence the intensity of its use.

**Table 3.** Estimated double hurdle model for factors influencing the adoption of SAI and intensity of use in Katete district, Zambia.

| Variables | Independent Double Hurdle Model | | | |
| --- | --- | --- | --- | --- |
| | 1st Hurdle (Decision to Adopt SAI) | Marginal Effect in Probit Model | | 2nd Hurdle (Intensity of SAI Use) |
| | Coefficient | Coefficient | SE | Coefficient |
| Constant | 1.570 ** (3.02) | - | - | 0.699 ** (4.80) |
| Number of people in the household | −0.041 (−0.79) | 0.0019 | 0.0079 | 0.015 (1.18) |
| Number of economically active adults | 0.073 (0.89) | 0.0059 | 0.0126 | −0.005 (−0.25) |
| Age of the household head in years | 0.004 (0.33) | −0.0008 | 0.0023 | −0.003 (−0.75) |
| Years in farming | −0.027 ** (−1.92) | −0.0016 | 0.0024 | 0.003 (0.71) |
| Ability to read and write (1 = yes; 0 otherwise) | −0.035 (−0.17) | −0.0046 | 0.0362 | −0.002 (0.02) |
| Number of work oxen owned | 0.065 (1.04) | −0.0041 | 0.0094 | −0.026 (−1.64) |
| Farm size (ha) | −0.047 (−1.14) | −0.0095 | 0.0050 | −0.010 ** (−2.01) |
| Total Cropped field size (ha) | −0.155 ** (-2.14) | −0.0231 | 0.0091 | −0.014 (−1.17) |
| Affiliation to farmer association (1 = yes; 0 otherwise) | 0.424 (1.73) | 0.1625 | 0.0455 | 0.227 ** (3.49) |
| Farmer extension training (1 = yes; 0 otherwise) | −0.290 (−0.55) | −0.1366 | 0.0678 | −0.278 ** (−2.09) |
| Received crop production training (1 = yes; 0 otherwise) | 0.308 (0.62) | 0.1516 | 0.0766 | 0.274 ** (2.18) |
| Cragg hurdle regression | | Number of observations | | 236 |
| | | LR chi2(9) | | 68.22 |
| | | Prob > chi2 | | 0.0000 |
| Log likelihood = −98.206077 | | Pseudo R2 | | 0.2578 |

Notes: ** represents 5% significance level. Variables are as defined in Equation (2). Figures in parentheses are z-values. Source: Estimated from 2020–2021 sample survey data.

The coefficients in the first hurdle (estimated via the probit model) indicate how a given decision variable affects the probability of adopting SAI. The results show that years of farming and crop area cultivated were the defining drivers for the decision to adopt SAI practices. The coefficient on years of farming is negative and significant (at 95% level of significance) indicating that generally, respondents who had been farming for some years were less likely to adopt SAI practices compared to respondents who were relatively new to farming. This could be a result of farmers with higher experience being more rigid or reluctant to change from practices they are used to compared to new entrants into farming starting with what is being promoted. Secondly, more years of farming could be correlated with older farmers who also have more land generally to expand. This finding is in harmony with other studies [44] that show that larger farms were more likely than smaller farms to expand. It is expected that adopters are likely to gain farming knowledge and expertise over time. The marginal effect shows that a very small increase in years of experience from the mean reduced the probability of adopting SAI practices by 0.0016 percentage points holding all other factors constant.

Similarly, crop area cultivated was negatively associated with the decision to adopt SAI practices. The coefficient on crop area cultivated was negative and significant, meaning that decreasing crop area cultivated led to a decrease in the probability of deciding to adopt

SAI practices by a given household. A small unit increase in crop area cultivated from the mean led to a 0.0237 percentage point reduction in the probability of deciding to adopt SAI practices holding all other factors constant. This finding is in harmony with other studies [44] that found a unit increase in farm size reduces the likelihood of adopting SAI practices that have a higher cost implication as the crop field size increases, such as the use of fertilizer as a soil fertility improvement practice. This result is also attributed to the low-income levels coupled with competing financial demands from other farm operations and inputs among the financially resource-constrained rural farm households. This result is also supported by other scholars [45] who assert that the consumption of fertilizer in Sub-Sahara Africa is very low and requires policy support in terms of subsidies and credit to make the input more affordable as they expand their crop area under production to meet the food security requirement. Farmers are also less likely to invest resources through the increasing area under long-term sustainable soil management practices such as green manure, cover crops, and agroforestry technologies due to competing immediate livelihood needs coupled with challenges of land tenure security challenges, especially under the customary land tenure systems [46,47].

The second hurdle of the double hurdle model estimated the drivers of the intensity of SAI adoption via the truncated regression model, which uses a maximum likelihood estimator to derive efficient and consistent model parameters. The results show that affiliation to a farmer group and crop production training received positively influenced the intensity of SAI adoption whereas increasing farm size, and farmer access to extension training negatively influenced the intensity of SAI adoption (Table 3).

The coefficient on farmer affiliation to association reveals that farmers belonging to an association had 0.2 more hectares planted under SAI than the farmers not affiliated to any association. This finding could be attributed to the role that farmer associations serve in Zambia, particularly as a conduit for farmer government and NGO-agricultural subsidized input support. Farmer associations also serve as a platform for information sharing and peer-to-peer learning among farmers, which influence the adoption of improved farming practices such as those related to SAI. This is also supported by other scholars [8,46], who indicate that within the wider community, mutual support networks and social groups contribute to the acquisition of knowledge, skills, and technologies that influence the choice of farming practices.

In contrast, the farm size of the household significantly and negatively affected the increasing intensity of SAI adoption. Precisely, increasing farm size owned by one hectare reduced the area under SAI practices by 0.01 hectares holding all other factors constant. This result corroborates the finding of other scholars [47,48]. Similar to farm size, access to extension services negatively influenced the increase in intensity of SAI practices. This result could be associated with farmer limitations in applying the SAI practices they are trained in by extension, hence leading to accessing an incomplete package of extension recommendations and hence the low area under SAI practices. Similar studies [49] show that despite farmers being very keen to make use of SAI practices such as fertilizers, improved seeds, and pesticides in their fields, limited access to the inputs coupled with the high cost limits the area under production for such practices despite extension efforts to educate farmers on their benefits. Households accessing extension services had 0.4 hectares of land under SAI compared to those who did not have access to extension services. Results also show that farmer training in crop production significantly impacted the intensity of adoption of SAI practices. Specifically, increasing farmer access to crop production training increased the area under SAI practices by 0.15 hectares. This finding is supported by other studies in sustainable and conservation agriculture [50], which indicate that farmers who have access to agricultural extension training that includes crop production tend to have a significantly larger area under sustainable agricultural practices than those who do not have access to extension.

## 5. Discussion

We set out to determine the factors that influence farmers' decisions to adopt sustainable agriculture intensification practices as well as those factors that influence farmers' decisions to increase or decrease area under SAI practices. Our findings revealed the importance of social, human, and natural capital in influencing SAI-related farming practices adoption. This finding is supported by [51], who also emphasized that understanding the financial, natural, physical, and social capital required to execute climate-smart agriculture technological practices are keys to their adoption. In particular, literacy level, farmer affiliation with farmer associations, total farmland owned, and cultivated area, as well as farm access to agricultural extension training, were found to positively influence the adoption of SAI-related practices. This could be attributed to the fact that the adoption of SAI practices is viewed positively as a means of increasing productivity. Adopters of SAI had higher literacy levels than non-adopters. In addition, the adopters showed a higher level of affiliation to farmer associations compared to non-adopters. This finding could be attributed to the assertion that literacy through formal as well as informal methods is an important element of human development and is a prerequisite to knowledge and the ability to apply formal skills, including the adoption of technology and other improved agricultural practices [52]. The finding is further supported by other studies [53,54], which indicate that human resource development is central to technology adoption and promotion of sustainable development and the alleviation of poverty. Therefore, the importance of farmer literacy in influencing technology adoption cannot be discounted. Studies by [55,56] have previously articulated that as farmer education levels rise, their willingness to take up new technologies increases because their levels of awareness as well as possible commitments to invest in such technologies increase.

Meanwhile, farmers' belonging to a farmer association has a critical role in shaping farmer behavior as a result of positive social group dynamics as well as routine exchanges of information. This demonstrates clearly the importance of social capital among rural communities in Africa. This has previously been reported by [46] in a research study conducted in Hoima district of Uganda. It is known that for most of Sub-Saharan Africa, agricultural extension information is transmitted largely through farmer-to-farmer exchanges [56–58]. Furthermore, government interventions in Zambia have sought to reach out to farmers that are organized in farmer groups as a way of enriching learning and social accountability [59]. It is thus not surprising that these results have revealed a positive influence of farmer associations and are an important demonstration of why governments need to further strengthen investment in ensuring proper and timely access to extension services. Evidence from Uganda [60] and a systematic review of literature by [61] have also shown that extension services play a central role in the adoption of climate-smart agriculture technologies that impact positively on boosting agricultural productivity among poor farmers through improved knowledge, education, and information, which farmers gain through agricultural extension services. Additionally, [62] has articulated that extension services were at the heart of the success of the green revolution through support and promotion of agricultural intensification in the small-scale and family farm sector in east Asia.

Smaller farm size was found to negatively impact the intensity of SAI adoption. This finding could be attributed to the need for adequate land that is necessary for sustaining the application of different practices aimed at increasing crop productivity without posing land constraints. The result is in agreement with other studies [10,46]. Additional research has also maintained this finding and asserts that agricultural expansion in Zambia is primarily driven by declining crop productivity due to declining soil fertility as a result of poor agronomic practices on existing pieces of land. Therefore, farmers tend to expand their land into virgin areas to compensate for low productivity in small, and in some cases old fields [63–65]. Farmer affiliation to farmer associations was significant in influencing the adoption intensity of SAI. This could be attributed to the reality that farmer associations in Zambia, particularly farmer cooperatives, have been a key conduit for government-supported subsidized agricultural inputs that include: improved crop varieties, herbicides,

pesticides, and fertilizers. This is also supported by other scholars [8,46] who indicate that social capital, particularly affiliation to farmer cooperatives, has been crucial in supporting the goal of increasing crop productivity. Existing studies further assert that increasing the complementarity between extension providers and encouraging changes in extension approaches supports the finding that extension worker training is critical for the sustainability of agriculture and environmental issues [40]. Literature on the subject additionally maintains that the type of extension training farmers receive is also key in helping farmers in decision-making pertaining to the application of different sustainable agricultural practices that suit their environment [42,62].

## 6. Conclusions

Our study focused on determining the factors that influence the decisions to adopt and the adoption intensity of SAI practices in Zambia. We find that adoption factors are those aligned to increasing the productivity of the farmer's field. In particular, sociocultural and human capital factors played a key role. In this regard, they underpin the importance of agricultural extension, farmer associations–farmer groups and cooperatives, and farmers' education in driving the adoption of SAI in Zambia. It is important to note that farmers that had a higher cropped area equally had higher adoption levels, revealing the fundamental action of opportunity cost in farmer decision-making: because of the level of their initial investment, they are willing to make additional investments to secure their cropped land. In terms of adoption intensity of SAI, the decision variables were influenced by farm size, affiliation to farmer association, farmer extension training, and type of extension training received. On the basis of these findings, we recommend increasing investment in agricultural extension and supporting farmer groups, associations, and cooperatives with increased training to strengthen their operational capacity as well as overall literacy of farmers with regards to SAI as a package of working practices.

**Author Contributions:** This research was successfully done with critical contributions from the authors. P.H. played a key role in data capturing and drafting of the manuscript. G.K. and E.K. supervised and guided the research process in terms of reviewing the data collection tools and methodology. A.E. was the main reviewer of the initial drafts and supported the harmonization of the manuscript in terms of structuring. R.A. contributed to the econometric analysis for the study. All authors have read and agreed to the published version of the manuscript.

**Funding:** This research was funded by UK Research & Innovation (UKRI) through the Global Challenges Research Fund (GCRF) program, Grant Ref: ES/P011306/under the project Social and Environmental Trade-offs in African Agriculture (SENTINEL) led by the International Institute for Environment & Development (IIED) in part implemented by the Regional Universities Forum for Capacity Building in Agriculture (RUFORUM).

**Informed Consent Statement:** Informed consent was obtained from all subjects involved in the study.

**Data Availability Statement:** Data will be made available upon request from the authors.

**Acknowledgments:** The fieldwork for this study drew a lot of logistical support from the Zambia-based Social and Environmental Trade-offs in African Agriculture (Sentinel) project. We also wish to sincerely thank Jacob Mwitwa of Copperbelt University, who coordinated the Sentinel project in Zambia and administratively supported the research process for this study.

**Conflicts of Interest:** The authors declare no conflict of interest. The funders had no role in the design of the study; in the collection, analyses, or interpretation of data; in the writing of the manuscript, or in the decision to publish the results.

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
