# Peer review of "Adoption of Sustainable Agriculture Intensification in Maize-Based Farming Systems of Katete District in Zambia"

_land, doi:10.3390/land11060880_

Round 1
Reviewer 1 Report
This is an important study and makes effective contribution to the literature on sustainable agricultural intensification. However, there is a need for the introduction to be re-written for clarity and brevity. It tends to contain some repetition in its current form. For example, the issue of soil nutrient depletion is not properly presented. The link between SSA and Zambia which is the specific study area in the introduction should be well established such that the introduction is more coherent. There seems to be more emphasis on SSA in the introduction than on Zambia and we have a situation where the authors focused initially on SSA, then to Zambia and back to SSA. The authors should ensure that the format of the paper is in line with the journal requirements. For example the sub-headings should be in italics and properly labeled. The Tables as well, should be in line with the journal format. More importantly, there is a need for the results of the econometric analysis to be better expressed with simplicity and clarity. Other minor points are highlighted below:
Abstract
There should be “as” after hailed in line 11
The word “adoption” after SAI in line 13 should be deleted
Introduction
In line 30 there should be “for” before “about”
The sentence between line 39 and line 42 should be refined for clarity
The phrase land “degradation due to land use land cover change” in line 58 should be checked for clarity
The word “agricultural” expansion in line 71 should be replaced with “land degradation”
There should be “of” after intensity in line 87
Methodology
The source of Figure 1 should be stated. If it is the authors’ compilation, that should be stated
In the household sampling, appropriate references with dates should be provided for the information stated within the subsection. For example “The Lukweta 104 community has 147 villages with an estimated farm household population of over 6,500 families.”
The word “estimated” in line 116 should be deleted
The exact number of research assistants involved in the data collection should be stated (line 123)
In line 170, “a vector of the error term” not “error terms”
Results
The results Table should be better presented in line with the journal requirements
The sentence in line 197 to 198 should be better expressed
The sentences in line 207 to 210 should be reframed especially in respect of the use of the word “popularity”
The mode of expression of the results should be improved upon. For example, the negative relation between adoption of SAI and years of farming should simply be expressed as, “the probability of adoption of SAI decreases with increasing years of farming” The statement “decreasing years of farming” is not the right way to go about it. You cannot decrease years of farming.
Line 265 “double” not doube”
Author Response
Please find attached the comprehensive point by point response to your comments and suggestions.

Reviewer 2 Report
This study aimed to examine the factors that determine farmers’ adoption of sustainable agricultural practices and their intensity in Zambia. Overall, it is an interesting paper. However, more needs to be done to bring it to a publishable level. In particular, the literature review and study area parts need to be strengthened. A theoretical framework is also needed to provide a sharper analytical focus. Following are other comments.
- The ‘study area’ part is very thin. More contextual information (e.g., economic and demographic characteristics) are needed to make it more nuanced.
- The variables used in the probit model lack theoretical justification. All variables need to be justified as to why they are appropriate. For instance, why is the age of the farmer important? Similarly, why is farming experience important? The authors should avoid the implicit assumption that readers know or can relate to the relevance of the selected variables.
- The authors reported p-values in Table 1, but it is unclear which statistical test was conducted. This must be clarified.
- Study limitations would be welcome.
Author Response
Please find attached the point by point response to your comments and suggestions

Round 2
Reviewer 1 Report
The authors have adequately addresses the issues I raised and the paper has been greatly improved.